# Fair Infinitesimal Jackknife: Mitigating the Influence of Biased Training Data Points Without Refitting

**Prasanna Sattigeri**
IBM Research
Yorktown Heights, NY
psattig@us.ibm.com

**Soumya Ghosh**
MIT-IBM Watson AI Lab, IBM Research
Cambridge, MA
ghoshso@us.ibm.com

**Inkit Padhi**
IBM Research
Yorktown Heights, NY
inkit.padhi@ibm.com

**Pierre Dognin**
IBM Research
Yorktown Heights, NY
pdognin@us.ibm.com

**Kush R. Varshney**
IBM Research
Yorktown Heights, NY
krvarshn@us.ibm.com

## Abstract

In consequential decision-making applications, mitigating unwanted biases in machine learning models that yield systematic disadvantage to members of groups delineated by sensitive attributes such as race and gender is one key intervention to strive for equity. Focusing on demographic parity and equality of opportunity, in this paper we propose an algorithm that improves the fairness of a pre-trained classifier by simply dropping carefully selected training data points. We select instances based on their influence on the fairness metric of interest, computed using an infinitesimal jackknife-based approach. The dropping of training points is done in principle, but in practice does not require the model to be refit. Crucially, we find that such an intervention does not substantially reduce the predictive performance of the model but drastically improves the fairness metric. Through careful experiments, we evaluate the effectiveness of the proposed approach on diverse tasks and find that it consistently improves upon existing alternatives.

## 1 Introduction

Among the many possible interventions to improve equity in society (most of them involve structural policy change), bias mitigation algorithms constitute one narrow sliver that has emerged in the machine learning literature to address distributive justice in high-stakes automated decision making. These algorithms may be categorized into pre-processing, in-processing, and post-processing approaches [37]. In the case of in-processing algorithms [20], the bias mitigation intervention occurs at the model training stage. This is usually achieved by minimizing the empirical risk regularized by a fairness metric surrogate that captures the dependence of the prediction and the sensitive attribute. Pre-processing methods typically learn transformations of the data distribution such that they do not contain information about the sensitive attributes [43, 26]. Task specific models are then learned from scratch on these debiased representations. Retraining a model from scratch is intractable in many real-world situations for a variety of reasons including policy, cost, and technical feasibility; post-processing approaches are the only viable option in such cases. For example, consider trying to refit large foundation models. Limiting ourselves to notions of *group* fairness such as demographic parity and equality of opportunity, existing post-processing bias mitigation algorithms tend to either randomly or deterministically alter the hard or soft predicted label of individual test data points that have been scored by a model [19, 17, 30, 8, 25, 39].

36th Conference on Neural Information Processing Systems (NeurIPS 2022).

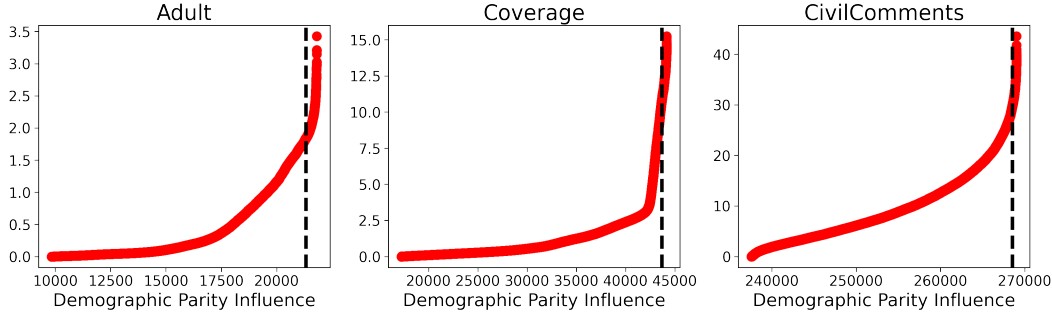

Figure 1: **Fairness influence**. We plot the sorted influence scores of training instances on the average demographic parity of a held-out validation set for different datasets considered in this paper. Only training instances with positive influence on demographic parity are plotted. The points to the right of the black line are the $500$ most influential training instances. Models adjusted to mitigate the influence of these instances are substantially more fair. On held-out test sets of the *Adult*, *Coverage*, *CivilComments* datasets, demographic parity improves from $0.18$ to $0.01$, from $0.22$ to $0.01$, and $0.17$ to $0.03$.

In this paper, we propose a more "global" bias mitigation algorithm. Our procedure alters the entire model without a focus on individual *test* points. Similar to post-processing approaches, our method mitigates pre-trained models without requiring any additional refitting of the model. Unlike standard post-processing approaches, however, our method does require access to the training data. In exchange for this additional requirement, we find that our approach typically substantially outperforms other post-processing techniques and can even augment in-processing approaches for a better fairness/accuracy trade-off.

**Our contributions.** Our first contribution is methodological. We use the notion of influence functions to estimate the "influence" of training instances on various group fairness metrics of interest. We then perform post-hoc unfairness mitigation by approximately removing training instances that have a disproportional impact on group (un)fairness. We theoretically analyze the proposed approach and establish conditions under which it provably improves group fairness.

Next, we observe that influence calculations require the inversion of a Hessian matrix, a prohibitively expensive operation for models with a large number of parameters. Existing approximations [21, 1, 34] can either be expensive, inaccurate, or unstable [29, 4, 35]. We develop *IHVP-WoodFisher*, a WoodFisher [35] based Inverse-Hessian Vector Product (IHVP) scheme for computing the fairness influence score of the training instances that is stable, easy to compute, and does not require constraints, such as restricted Eigenspectrum of the loss curvature, that are hard to satisfy in practice.

Our final contribution is empirical. First, through careful experiments on tabular data, we show that our approach is effective at reducing group unfairness, is competitive with existing methods, and can even augment the latter to achieve a better fairness/accuracy trade-off. Then, we demonstrate how our approach can be easily adapted to more complex modalities such as natural language and be used for bias mitigation of large pre-trained language models through *prompt-tuning*, a use-case that is likely to become increasingly common with the proliferation of large language models.

## 2  Background and Related Work

### 2.1  Empirical and Weighted Risk Minimization

We begin by considering the standard supervised learning setup. Given a dataset $\mathcal{D} = \{\mathbf{z}_n = (\mathbf{x}_n, y_n)\}_{n=1}^{N}$ of $N$ features ($\mathbf{x}_n \in \mathbb{R}^p$), response pairs ($y_n \in \mathcal{Y}$), a model $h_{\boldsymbol{\theta}}(\mathbf{x})$ parameterized by a set of parameters $\boldsymbol{\theta} \in \Theta \subseteq \mathbb{R}^D$, and a loss function $\ell : \Theta \times \mathcal{Y} \to \mathbb{R}$, we minimize the empirical risk,

$$\hat{\boldsymbol{\theta}} = \underset{\boldsymbol{\theta} \in \Theta}{\operatorname{argmin}} \frac{1}{N} \sum_{n=1}^{N} \ell(y_n, h_{\boldsymbol{\theta}}(\mathbf{x}_n)), \tag{1}$$

to arrive at a trained predictor $h_{\hat{\boldsymbol{\theta}}}(\mathbf{x})$. We will denote $L(\boldsymbol{\theta}) \stackrel{\text{def}}{=} \frac{1}{N} \sum_{n=1}^{N} \ell(y_n, h_{\boldsymbol{\theta}}(\mathbf{x}_n))$ for notational convenience. Next, consider a weighted risk minimization problem,

$$\hat{\boldsymbol{\theta}}(\mathbf{w}) = \operatorname*{argmin}_{\boldsymbol{\theta} \in \Theta} \frac{1}{N} \sum_{n=1}^{N} w_n \ell(y_n, h_{\boldsymbol{\theta}}(\mathbf{x}_n)), \tag{2}$$

that weights the loss at each training instance by a scalar weight $w_n$ and $\mathbf{w}$ denotes the column vector $[w_1, w_2, \ldots, w_N]^T \in \mathbb{R}^N$. Setting all the weights to one, $\mathbf{1} \stackrel{\text{def}}{=} [w_1 = 1, w_2 = 1, \ldots, w_N = 1]^T$, and minimizing the right hand side of Equation 2 recovers the standard empirical risk minimization problem. On the other hand, setting the $n^{\text{th}}$ coordinate to zero recovers the solution to an empirical risk minimization problem after dropping the $n^{\text{th}}$ training instance. As is clear from Equation 2 and emphasized by our notation, $\boldsymbol{\theta}$ is a function of the weights $\mathbf{w}$. Although we typically do not have a closed form expression for this function, we can form a Taylor approximation to it:

$$\boldsymbol{\theta}(\mathbf{w}) = \hat{\boldsymbol{\theta}} + \nabla_{\mathbf{w}} \boldsymbol{\theta}(\mathbf{w}) \Big|_{\mathbf{w}=\mathbf{1}} (\mathbf{w} - \mathbf{1}) + \mathcal{O}((\mathbf{w} - \mathbf{1})^2), \tag{3}$$

where $\nabla_{\mathbf{w}} \boldsymbol{\theta}(\mathbf{w}) \in \mathbb{R}^{D \times N}$ is the Jacobian matrix. This first order Taylor approximation is often referred to as the infinitesimal jackknife approximation [18, 14]. The coordinate-wise gradient $\frac{d\theta(\mathbf{w})}{dw_n}|_{w_n=1}$ measures the effect of perturbing the weight of the $n^{\text{th}}$ data point on $\hat{\boldsymbol{\theta}}$ and is popularly referred to as the influence function [21], since it measures the "influence" of the $n^{\text{th}}$ training instance on the model's parameters. When we are at a stationary point of $L(\boldsymbol{\theta})$, i.e., when $\nabla_{\boldsymbol{\theta}} L(\boldsymbol{\theta}) = 0$, $L(\boldsymbol{\theta})$ is twice differentiable in $\boldsymbol{\theta}$, then,

$$\frac{d\theta(\mathbf{w})}{dw_n} \Big|_{\mathbf{w}=\mathbf{1}} = -\mathbf{H}^{-1} \mathbf{g}_n, \tag{4}$$

where $\mathbf{H} \stackrel{\text{def}}{=} \nabla_{\boldsymbol{\theta}}^2 L(\boldsymbol{\theta})|_{\boldsymbol{\theta}=\hat{\boldsymbol{\theta}}}$, and $\mathbf{g}_n \stackrel{\text{def}}{=} \nabla_{\boldsymbol{\theta}} \ell(y_n, h_{\boldsymbol{\theta}}(\mathbf{x}_n))|_{\boldsymbol{\theta}=\hat{\boldsymbol{\theta}}}$. Recent work [15] has shown that the above expression approximates the gradient well in the vicinity of a stationary point with the accuracy of the approximation deteriorating smoothly with increasing distance from the stationary point. This result justifies the use of influence functions even when stochastic optimization is used for minimizing Equation 1. Finally, to measure the influence of a training instance on a differentiable functional, $M$, of $\boldsymbol{\theta}(\mathbf{w})$, we apply chain rule to arrive at,

$$\mathcal{I}_{M,n} \stackrel{\text{def}}{=} \frac{dM(\boldsymbol{\theta}(\mathbf{w}), \mathbf{w})}{dw_n} \Big|_{\mathbf{w}=\mathbf{1}, \boldsymbol{\theta}=\hat{\boldsymbol{\theta}}} = -\nabla_{\boldsymbol{\theta}} M(\boldsymbol{\theta}(\mathbf{w}), \mathbf{w}) \Big|_{\mathbf{w}=\mathbf{1}, \boldsymbol{\theta}=\hat{\boldsymbol{\theta}}}^T \mathbf{H}^{-1} \mathbf{g}_n, \tag{5}$$

where our notation makes explicit the dependence of $M$ on $\mathbf{w}$. Recent work has leveraged this machinery to approximate cross-validation [16, 36, 15], to interpret black-box machine learning models [21], and to assess the sensitivity of statistical analyses to training data perturbations [6], among others. Differently from these, we show how this machinery can be leveraged for reducing disparities of pre-trained models across groups.

## 2.2 Fair Classification

We further assume that for each data instance we have access to a sensitive attribute $\mathbf{s}_n \in [k]$, i.e., $\mathcal{D} = \{\mathbf{z}_n = (\mathbf{x}_n, s_n, y_n)\}_{n=1}^N$, that encodes the protected group membership of the $n^{\text{th}}$ data instance and that we are interested in binary classification, $\mathcal{Y} = \{1, 0\}$. In fair classification, we want to learn accurate classifiers that minimize disparities in predictions across groups.

To quantify disparities across groups, we primarily focus on two common fairness metrics — demographic (or statistical) parity (DP) [3] and equality of odds (EO) [17]. DP requires the classifier's predictions to be statistically independent of the sensitive attribute, $h_{\boldsymbol{\theta}}(X) \perp\!\!\!\perp S$, where $X$ and $S$ are random variables representing the features and the sensitive attribute. For a binary sensitive attribute, DP implies, $P(h_{\boldsymbol{\theta}}(X) = 1 \mid S = 1) = P(h_{\boldsymbol{\theta}}(X) = 1 \mid S = 0)$. EO, on the other hand, requires the classifier's predictions to be statistically independent of the sensitive attribute *conditioned* on the true outcome, $h_{\boldsymbol{\theta}}(X) \perp\!\!\!\perp S \mid Y$. For a binary sensitive attribute, EO implies $P(h_{\boldsymbol{\theta}}(X) = 1 \mid S = 1, Y = y) = P(h_{\boldsymbol{\theta}}(X) = 1 \mid S = 0, Y = y)$ for both $y = 0$ and $y = 1$. Equality of opportunity (EQOPP) [17] is a special case of equality of odds where the predictions are conditionally independent of the sensitive attribute given the true outcome is positive. A common

strategy for learning fair classifiers is to then require the absolute difference in demographic parity (DP),

$$\Delta\text{DP}(\boldsymbol{\theta}) = |P(h_{\boldsymbol{\theta}}(X) = 1 \mid S = 1) - P(h_{\boldsymbol{\theta}}(X) = 1 \mid S = 0)|,$$

or the absolute difference in equality of odds,

$$\Delta\text{EO}(\boldsymbol{\theta}) = \sum_{y=0}^{1} |P(h_{\boldsymbol{\theta}}(X) = 1 \mid S = 1, Y = y) - P(h_{\boldsymbol{\theta}}(X) = 1 \mid S = 0, Y = y)|,$$

to be close to zero while minimizing the empirical risk (Equation 1). Smooth[1] surrogates to $\Delta\text{DP}$ and $\Delta\text{EO}$ that are estimated from an empirical distribution are commonly used in practice [42],

$$M_{\mathcal{D}}^{\Delta\text{DP}}(\boldsymbol{\theta}) = |\mathbb{E}_{p_{\mathcal{D}}(X=\mathbf{x}|S=1)}[h_{\boldsymbol{\theta}}(\mathbf{x})] - \mathbb{E}_{p_{\mathcal{D}}(X=\mathbf{x}|S=0)}[h_{\boldsymbol{\theta}}(\mathbf{x})]|$$

$$M_{\mathcal{D}}^{\Delta\text{EO}}(\boldsymbol{\theta}) = \sum_{y=0}^{1} |\mathbb{E}_{p_{\mathcal{D}}(X=\mathbf{x}|S=0,Y=y)}[h_{\boldsymbol{\theta}}(\mathbf{x})] - \mathbb{E}_{p_{\mathcal{D}}(X=\mathbf{x}|S=1,Y=y)}[h_{\boldsymbol{\theta}}(\mathbf{x})]|, \tag{6}$$

where $M_a^b(\boldsymbol{\theta})$ denotes the surrogate for the fairness metric $b$ estimated from dataset $a$. When $\boldsymbol{\theta}$ is itself a function of $\mathbf{w}$, we will use the notation $M_a^b(\boldsymbol{\theta}(\mathbf{w}), \mathbf{w})$.

### 2.3 Other Related Work

Many pre-processing based bias mitigation algorithms, learn low dimensional representations of the data that are independent of the sensitive attribute [43, 26]. Others aim to learn fairness promoting transformations in the ambient space of the data [7, 32, 40]. Pre-processing methods that transform the data points can often run the risk of losing the semantics of the original data points. Often, they can be expensive, especially for high-dimensional data and large datasets. Furthermore, they must be performed before training any task-specific models and thus are not applicable when the goal is to improve a model already trained with an expensive procedure. In [38], the authors first obtain a counterfactual feature distribution by identifying the test instances, which when dropped the pre-trained model predictions are fair on the remaining test instances. They then learn a optimal transport based randomized pre-processor that maps the transforms the new test samples from the unprivileged group to fair counterfactual distribution. In contrast, our goal is to compute the influence scores for the training instances, which is more challenging. Additionally, [38] requires the sensitive attributes be known at test time as the pre-processor is specific to the unprivileged group. Instead, we aim to directly edit the trained model and eliminate the need of sensitive attribute labels at test time.

In-processing algorithmic fairness methodologies [7, 20] are applicable when we can train models along with fairness constraints. Mary et al. [27] enforce independence through a relaxation of the Hirschfeld-Gebelein-Rényi Maximum Correlation Coefficient (HGR) dependency measure. Similarly, *Rebias* [2, 41] uses the Hilbert-Schmidt Independence Criterion (HSIC) to reduce the dependence of the representations on the sensitive attibutes. *FairMixup* [9] is a data augmentation strategy to improve the generalization properties of in-processing algorithms. These methods can be sensitive to the regularization strength and can sacrifice too much accuracy. In contrast, our approach is applicable when the base model to trained unconstrained on the main task, which can then be updated to remove the influence of the harmful instances and improve fairness.

Existing post-processing methods [19, 17, 30, 25, 39, 41] learn to transform the predictions of a trained model to satisfy a measure of fairness. These can often be limiting as they do not provide control over the fairness accuracy trade-off, may require that predicted scores to be well-calibrated, or may lead to excessive reduction in performance. In contrast, our method exploits the training data and model gradients efficiently to generate stronger, yet computationally inexpensive post-hoc interventions at minimal loss of predictive performance.

## 3 Fair Classification through Post-Hoc Interventions

We now develop and analyze a post-processing fairness algorithm that given (i) a pre-trained model, (ii) access to the training data and optionally a validation set, (iii) a twice differentiable loss function

---

[1]Nearly smooth. The absolute value is not differentiable at zero, but this is not a concern since we rarely encounter exact zeros in practice.

and a once differentiable surrogate to the fairness metric, and (iv) an invertible Hessian at a local optimum of the loss, improves the fairness characteristics of a pre-trained model without requiring it to be refit.

## 3.1 Influence Functions for Group Fairness

Assuming that we use a held-out validation set $\mathcal{D}_{\text{val}} = \{\mathbf{x}_n, s_n, y_n\}_{n=1}^{N_{\text{val}}}$ to estimate $M_{\mathcal{D}_{\text{val}}}^{\Delta\text{DP}}(\boldsymbol{\theta})$ and $M_{\mathcal{D}_{\text{val}}}^{\Delta\text{EO}}(\boldsymbol{\theta})$, we can leverage the result in Equation 5 to compute the influence of the $n^{\text{th}}$ training instance on $\Delta\text{DP}$,

$$
\begin{aligned}
\mathcal{I}_{\Delta\text{DP},n} &= -\nabla_{\boldsymbol{\theta}} M_{\mathcal{D}_{\text{val}}}^{\Delta\text{DP}}(\hat{\boldsymbol{\theta}})^T \mathbf{H}^{-1} \mathbf{g}_n, \\
&= -\nabla_{\boldsymbol{\theta}} | \mathbb{E}_{p_{\mathcal{D}}(X=\mathbf{x}|S=1)}[h_{\boldsymbol{\theta}}(\mathbf{x})] - \mathbb{E}_{p_{\mathcal{D}}(X=\mathbf{x}|S=0)}[h_{\boldsymbol{\theta}}(\mathbf{x})]| \big|_{\boldsymbol{\theta}=\hat{\boldsymbol{\theta}}} \mathbf{H}^{-1} \mathbf{g}_n,
\end{aligned}
\tag{7}
$$

and on $\Delta\text{EO}$,

$$
\begin{aligned}
\mathcal{I}_{\Delta\text{EO},n} &= -\nabla_{\boldsymbol{\theta}} M_{\mathcal{D}_{\text{val}}}^{\Delta\text{EO}}(\hat{\boldsymbol{\theta}})^T \mathbf{H}^{-1} \mathbf{g}_n, \\
&= -\nabla_{\boldsymbol{\theta}} | \sum_{y=0}^{1} \mathbb{E}_{p_{\mathcal{D}}(X=\mathbf{x}|S=0,Y=y)}[h_{\boldsymbol{\theta}}(\mathbf{x})] - \mathbb{E}_{p_{\mathcal{D}}(X=\mathbf{x}|S=1,Y=y)}[h_{\boldsymbol{\theta}}(\mathbf{x})] | \Big|_{\boldsymbol{\theta}=\hat{\boldsymbol{\theta}}} \mathbf{H}^{-1} \mathbf{g}_n,
\end{aligned}
\tag{8}
$$

where we have used $M_{\mathcal{D}_{\text{val}}}^{\Delta\text{DP}}(\hat{\boldsymbol{\theta}})$ and $M_{\mathcal{D}_{\text{val}}}^{\Delta\text{EO}}(\hat{\boldsymbol{\theta}})$ to denote $M_{\mathcal{D}_{\text{val}}}^{\Delta\text{DP}}(\boldsymbol{\theta}(\mathbf{w}), \mathbf{w})\big|_{\mathbf{w}=\mathbf{1}, \boldsymbol{\theta}=\hat{\boldsymbol{\theta}}}$ and $M_{\mathcal{D}_{\text{val}}}^{\Delta\text{EO}}(\boldsymbol{\theta}(\mathbf{w}), \mathbf{w})\big|_{\mathbf{w}=\mathbf{1}, \boldsymbol{\theta}=\hat{\boldsymbol{\theta}}}$. We highlight that computing the influence of training instances on group fairness metrics requires solving a *single* empirical risk minimization problem to recover $\hat{\boldsymbol{\theta}}$. The fairness metrics could also be estimated on the training data if no validation set is available. However, empirically we find that a validation set improves results.

## 3.2 Post-Hoc Mitigation

Revisiting Equation 3, we note that the first order Taylor approximation about $\mathbf{1}$ is a function of $\mathbf{w}$. This opens up the possibility of post-hoc fairness improvement of a pre-trained $\hat{\boldsymbol{\theta}}$ by searching for a weight vector $\mathbf{w}_{\text{fair}}$ such that $M_a^b(\hat{\boldsymbol{\theta}}_{\text{fair}}) \approx 0$, where,

$$
\begin{aligned}
\hat{\boldsymbol{\theta}}_{\text{fair}} &\stackrel{\text{def}}{=} \hat{\boldsymbol{\theta}}(\mathbf{w}_{\text{fair}}) = \hat{\boldsymbol{\theta}} + \sum_{n=1}^{N} \frac{d\boldsymbol{\theta}(\mathbf{w})}{dw_n}\bigg|_{\mathbf{w}=\mathbf{1}} (w_n^{\text{fair}} - 1), \\
&= \hat{\boldsymbol{\theta}} - \sum_{n=1}^{N} \mathbf{H}^{-1} \mathbf{g}_n (w_n^{\text{fair}} - 1),
\end{aligned}
\tag{9}
$$

and $\mathbf{w}_{\text{fair}} = [w_1^{\text{fair}}, w_2^{\text{fair}}, \ldots, w_N^{\text{fair}}]^T \in \mathbb{R}^N$. We could use gradient-based methods to learn $\mathbf{w}_{\text{fair}}$ by optimizing a desired $M(\boldsymbol{\theta}(\mathbf{w}), \mathbf{w})$ with respect to $\mathbf{w}$. However, computing and inverting the Hessian requires $\mathcal{O}(ND^2 + D^3)$ operations and is prohibitively expensive for large models. Instead, iterative procedures involving repeated Hessian-vector products are often used in practice [21]. A gradient-based procedure would need to either perform this iterative procedure after *every* gradient step or pre-compute $\sum_n \mathbf{H}^{-1} \mathbf{g}_n$, rendering the procedure computationally intractable for most cases of interest. Moreover, solely optimizing $M(\boldsymbol{\theta}(\mathbf{w}), \mathbf{w})$ will likely result in fair but inaccurate classifiers, and the optimized weights will typically not be interpretable.

We circumvent these issues by constraining the elements of $\mathbf{w}$ to be binary. In Proposition 3.1, we show that we can construct $\mathbf{w}_{\text{fair}}$ by simply zeroing out coordinates of $\mathbf{w}_{\text{fair}}$ that correspond to training instances with a positive influence on the fairness metric of interest. This construction is inherently interpretable. Setting an element to zero implies training without the corresponding training instance. Zeroing out instances with positive influence equates to refitting the model after dropping training instances that increase disparity across groups.

We now establish conditions under which $\mathbf{w}_{\text{fair}}$ as constructed above leads to classifiers with lower group disparities. Let $\mathbf{1} \in \mathbb{R}^N$ denote an $N$-dimensional vector of all ones, $b$ denote a fairness metric, $\overline{M}_{\mathcal{D}_{\text{val}}}^b(\boldsymbol{\theta}(\mathbf{w}), \mathbf{w})$ denote a linearized approximation to $M_{\mathcal{D}_{\text{val}}}^b(\boldsymbol{\theta}(\mathbf{w})), \mathbf{w})$ about $\mathbf{1}$, and $\mathbb{1}[\alpha > \beta]$ denote an indicator function that takes the value one if $\alpha > \beta$ is true and zero otherwise.

**Proposition 3.1.** *Let $\mathbf{w}_{\mathrm{fair}} \in \{0,1\}^N$ be a N dimensional binary vector such that its $n^{\mathrm{th}}$ coordinate is $w_n^{\mathrm{fair}} = 1 - \mathbb{1}[\mathcal{I}_{b,n} > 0]$, then,*

$$\mathbf{w}_{\mathrm{fair}} = \underset{\mathbf{w} \in \{0,1\}^N}{\operatorname{argmin}} \, \overline{M}_{\mathcal{D}_{\mathrm{val}}}^{b}(\boldsymbol{\theta}(\mathbf{w}), \mathbf{w}) - M_{\mathcal{D}_{\mathrm{val}}}^{b}(\boldsymbol{\theta}(\mathbf{1}), \mathbf{1}),$$

*and $\overline{M}_{\mathcal{D}_{\mathrm{val}}}^{b}(\boldsymbol{\theta}(\mathbf{w}_{\mathrm{fair}}), \mathbf{w}_{\mathrm{fair}}) - M_{\mathcal{D}_{\mathrm{val}}}^{b}(\boldsymbol{\theta}(\mathbf{1}), \mathbf{1}) \leq 0$.*

*Proof.* Denote $M_{\mathcal{D}_{\mathrm{val}}}^{b}(\hat{\boldsymbol{\theta}}) := M_{\mathcal{D}_{\mathrm{val}}}(\boldsymbol{\theta}(\mathbf{1}), \mathbf{1})$. From a first order Taylor approximation about $\mathbf{1}$, we have,

$$
\begin{aligned}
\overline{M}_{\mathcal{D}_{\mathrm{val}}}^{b}(\boldsymbol{\theta}(\mathbf{w}), \mathbf{w}) &= M_{\mathcal{D}_{\mathrm{val}}}^{b}(\hat{\boldsymbol{\theta}}) + \sum_{n=1}^{N} \frac{dM_{\mathcal{D}_{\mathrm{val}}}^{b}(\hat{\boldsymbol{\theta}}(\mathbf{w}), \mathbf{w})}{dw_n}\bigg|_{\mathbf{w}=\mathbf{1}, \boldsymbol{\theta}=\hat{\boldsymbol{\theta}}}(w_n - 1), \\
&= M_{\mathcal{D}_{\mathrm{val}}}^{b}(\hat{\boldsymbol{\theta}}) + \sum_{n=1}^{N} \mathcal{I}_{b,n}(w_n - 1).
\end{aligned}
\tag{10}
$$

Rearranging terms,

$$
\begin{aligned}
\overline{M}_{\mathcal{D}_{\mathrm{val}}}^{b}(\boldsymbol{\theta}(\mathbf{w}), \mathbf{w}) - M_{\mathcal{D}_{\mathrm{val}}}^{b}(\hat{\boldsymbol{\theta}}) &= \sum_{n=1}^{N} \mathcal{I}_{b,n}(w_n - 1) \\
&= \sum_{n=1}^{N} \mathbb{1}[\mathcal{I}_{b,n} > 0]\mathcal{I}_{b,n}(w_n - 1) + \sum_{n=1}^{N} \mathbb{1}[\mathcal{I}_{b,n} \leq 0]\mathcal{I}_{b,n}(w_n - 1).
\end{aligned}
\tag{11}
$$

Finally, the result follows from observing that $w_n \in \{0,1\}$ and noting that the first term can be either zero (when $w_n = 1$) or negative (when $w_n = 0$ and $\mathbb{1}[\mathcal{I}_{b,n} > 0]$) and the second term can be either zero (when $w_n = 1$) or positive (when $w_n = 0$ and $\mathbb{1}[\mathcal{I}_{b,n} \leq 0]$). $\mathbf{w}_{\mathrm{fair}}$ drives the second term to zero and sets the first term to the smallest value attainable by a binary $\mathbf{w}$. $\square$

It follows that $M_{\mathcal{D}_{\mathrm{val}}}^{b}(\boldsymbol{\theta}(\mathbf{w}_{\mathrm{fair}}), \mathbf{w}_{\mathrm{fair}}) \approx\leq M_{\mathcal{D}_{\mathrm{val}}}^{b}(\hat{\boldsymbol{\theta}}(\mathbf{1}), \mathbf{1})$, with the inequality holding when the linearization is accurate. Finally, defining $\mathcal{D}_{-} = \{\mathbf{z}_n \mid \mathbf{z}_n \in \mathcal{D} \text{ and } \mathcal{I}_{M,n} > 0\}$, we arrive at the post-hoc mitigated classifier by plugging in $\mathbf{w}_{\mathrm{fair}}$ from Proposition 3.1 in Equation 9.

$$\hat{\boldsymbol{\theta}}_{\mathrm{fair}} = \hat{\boldsymbol{\theta}} + \sum_{m \in \mathcal{D}_{-}} \mathbf{H}^{-1}\mathbf{g}_m. \tag{12}$$

In Appendix A we consider an alternate $\mathbf{w}_{\mathrm{fair}}$ that is guaranteed to decrease *both* the loss $\ell$ and the fairness metric on the validation set $\mathcal{D}_{\mathrm{val}}$. In early experiments, we did not see consistent benefits from using this alternate version and do not consider it further in this paper.

### 3.3 Practical Considerations

**Hessian computation and inversion.** The influence function computation involves computing and inverting the Hessian of the loss function on the training data. This requires $\mathcal{O}(ND^2 + D^3)$ operations. Both computing and storing the Hessian becomes prohibitively expensive for large models. While diagonal approximations to the Hessian are possible, they tend to be inaccurate. Instead, iterative methods based on the (truncated) Neumann expansion have been proposed in the past [21]. However, more recent work has found the Neumann approximation to be inaccurate, cf. [36, Appendix C] and prone to numerical issues when the eigenvalues of the Hessian fall outside the $[0, 1]$ interval. Motivated by these shortcomings, here we develop an alternative iterative procedure based on the recently proposed WoodFisher approximation [35].

The WoodFisher approximation provides us with the following recurrence relation for estimating the inverse of the Hessian:

$$\hat{\mathbf{H}}_{n+1}^{-1} = \hat{\mathbf{H}}_n^{-1} - \frac{\hat{\mathbf{H}}_n^{-1}\nabla_\theta \ell(y_{n+1}, h_{\boldsymbol{\theta}}(\mathbf{x}_{n+1}))\nabla_\theta \ell(y_{n+1}, h_{\boldsymbol{\theta}}(\mathbf{x}_{n+1})^\top \hat{\mathbf{H}}_n^{-1}}{N + \nabla_\theta \ell(y_{n+1}, h_{\boldsymbol{\theta}}(\mathbf{x}_{n+1}))^\top \hat{\mathbf{H}}_n^{-1}\nabla_\theta \ell(y_{n+1}, h_{\boldsymbol{\theta}}(\mathbf{x}_{n+1}))}, \tag{13}$$

with $\hat{\mathbf{H}}_0^{-1} = \lambda^{-1}I_D$, and $\lambda$ a small positive scalar.

For computing influence functions we only need to store the product of the inverse Hessian with a vector $\mathbf{v}$, i.e., $\mathbf{H}^{-1}\mathbf{v}$, which should only require $\mathcal{O}(D)$ storage. However, if we first compute the inverse Hessian and then compute the Hessian-vector product (HVP), we would need $\mathcal{O}(D^2)$ storage. To sidestep this issue, we develop the following coupled recurrences that only use $\mathcal{O}(D)$ storage. We call these coupled recurrences *IHVP-WoodFisher*,

$$\boldsymbol{o}_{n+1} = \boldsymbol{o}_n - \frac{\boldsymbol{o}_n \nabla_\theta \ell(\mathbf{z}_{n+1})^\top \boldsymbol{o}_n}{N + \nabla_\theta \ell(\mathbf{z}_{n+1})^\top \boldsymbol{o}_n}, \quad \boldsymbol{k}_{n+1} = \boldsymbol{k}_n - \frac{\boldsymbol{o}_n \nabla_\theta \ell(\mathbf{z}_{n+1})^\top \boldsymbol{k}_n}{N + \nabla_\theta \ell(\mathbf{z}_{n+1})^\top \boldsymbol{o}_n}, \qquad (14)$$

where, we use $\ell(\mathbf{z}_{n+1})$ as shorthand for $\ell(y_{n+1}, h_\theta(\mathbf{x}_{n+1}))$, $\boldsymbol{o}_1 = \nabla_\theta \ell(y_1, h_\theta(\mathbf{x}_1))$, and $\boldsymbol{k}_1 = \mathbf{v}$.

**Proposition 3.2.** *Let $\boldsymbol{o}_1 = \nabla_\theta \ell(\mathbf{z}_1)$, $\boldsymbol{k}_1 = \mathbf{v}$, and $N$ denote the number of training instances. The Hessian-vector product $\mathbf{H}^{-1}\mathbf{v}$ is approximated by iterating through the IHVP-WoodFisher recurrence in Equation 14 and computing $\boldsymbol{k}_N$.*

We prove Proposition 3.2 in Appendix B. In practice, we observe that even using $B \ll N$ iterations produces useful approximations. In Appendix C, we compare the approximation accuracy of the the *IHVP-WoodFisher* and the iterative *Neumann* approach on cases where it is tractable to exactly compute the IHVP. Algorithm 2 (see Appendix) summarizes our vanilla approach.

**Computational speedups:** Although Algorithm 2 suggests running the *IHVP-WoodFisher* iterations for each training instance for clarity of exposition, in practice, we use the following trick to run the *IHVP-WoodFisher* iterations only once for the entire training dataset. First, for any $p * p$ symmetric matrix $\mathbf{A}$ and $p$-dimensional vectors $\mathbf{x}$ and $\mathbf{y}$, $\mathbf{x}^T \mathbf{A}, \mathbf{y} = \mathbf{y}^T \mathbf{A}\mathbf{x}$. From Equation 5, the influence calculation involves computing $\nabla_\theta M(\hat{\theta}, \mathbf{1})^T \mathbf{H}^{-1}\mathbf{g}_n$ for all $n$ in the training dataset. Since is symmetric, we can equivalently compute $\mathbf{g}_n^T \mathbf{H}^{-1}\nabla_\theta M(\hat{\theta}, \mathbf{1})$. We can then run the *IHVP-WoodFisher* iterations to approximate $\mathbf{H}^{-1}\nabla_\theta M(\hat{\theta}, \mathbf{1})$. Crucially, we need to do this only once. With the approximation in hand, computing the per data influence requires a single dot product per data instance between $g_n$ and the *IHVP-WoodFisher* approximated $\mathbf{H}^{-1}\nabla_\theta M(\hat{\theta}, \mathbf{1})$. In contrast to other approaches to scaling up influence functions [33] our approach only requires the storage of a single $p$-dimensional vector. We call this more efficient version `Fair-IJ` and is summarized in Algorithm 1.

**Most influential instances.**    Our development and analysis depends on first order linear approximations of non-linear functions about $\mathbf{1}$. We expect the quality of these approximations to deteriorate further away from $\mathbf{1}$, i.e., with increasing number of instances dropped. See Theorem 1 in [6] for additional discussion on the quality of approximation. We find that instead of dropping all instances with positive influence, dropping the $k$ most influential instances yields better bias mitigation. We select the hyperparameter $k$ that results in the lowest (best) fairness score on the validation set. Additionally, [35] observed that the *WoodFisher* Hessian estimate $\hat{\mathbf{H}}$ differs from the true Hessian by a scaling factor, i.e $\hat{\mathbf{H}} \propto \mathbf{H}$. We select, from a pre-specified set, the scaling factor that minimizes the fairness score on the validation set. We then scale the *IHVP-WoodFisher* estimates using the selected scaling factor. See Appendix D.

## 4    Experiments

We first study our method on tabular datasets including the well-known Adult dataset [13] and the recently released ACSPublicCoverage [11] dataset. ACSPublicCoverage is one among a suite of datasets aimed to be larger alternatives to previously available fairness datasets. We then investigate our method on the text modality and larger pre-trained models using the CivilComments dataset [5].

### 4.1    Tabular Datasets

**Setup.** The task in the Adult dataset is to predict if a person has an income above a threshold. We use gender as the sensitive attribute. This dataset comes with a fixed test set. A random 33% of the training data is used as the validation set for each trial of the experiments. We follow the pre-processing steps from [27]. The task in the ACSPublicCoverage dataset is to predict if a person has public health insurance coverage. For our experiments, we only consider instances from the year 2014, from the state of California, and belonging to the white or black race. We consider race as the

**Algorithm 1** `Fair-IJ`

---

1: **Input:** Pre-trained model parameters $\hat{\boldsymbol{\theta}}$, training set $\mathcal{D}$, loss function $\ell$, a validation set $\mathcal{D}_{\text{val}}$ and a smooth surrogate to the fairness metric $b \in \{\Delta\text{DP}, \Delta\text{EO}\}$, $M_{\mathcal{D}_{\text{val}}}^b$.
2: **Calculate:** $\nabla_{\boldsymbol{\theta}} M(\hat{\boldsymbol{\theta}}, \mathbf{1})$ using Equation 7 or Equation 8.
3: **Calculate:** $\boldsymbol{r} = \mathbf{H}^{-1} \nabla_{\boldsymbol{\theta}} M(\hat{\boldsymbol{\theta}}, \mathbf{1})$ by setting $\boldsymbol{k}_1 = \nabla_{\boldsymbol{\theta}} M(\hat{\boldsymbol{\theta}}, \mathbf{1})$ and iterating through Equation 14 for B iterations.
4: **Calculate:** the fairness influence $\mathcal{I}_{b,n}$ of each training instance $\mathbf{z}_n$ on $\mathcal{D}_{\text{val}}$ by computing dot product between $g_n$ and $\boldsymbol{r}$.
5: **Construct:** the set $\mathcal{D}_-$ and denote its cardinality, $|\mathcal{D}_-| = K$.
6: **Initialize:** $\hat{\boldsymbol{\theta}}_{\text{fair}}^0 := \hat{\boldsymbol{\theta}}$
7: **for** $k \in [1, \dots, K]$ **do**
8:     **Construct:** $\mathcal{D}_-^k = \{\mathbf{z}_n \in \mathcal{D}_- \mid \mathcal{I}_{b,n} > \mathcal{I}_{b,(K-k)}\}$, where $\mathcal{I}_{b,(K-k)}$ denotes the $(K-k)^{\text{th}}$ order statistic of the influence scores $[\mathcal{I}_{b,1}, \dots, \mathcal{I}_{b,K}]$.
9:     **Calculate:** $\hat{\boldsymbol{\theta}}_{\text{fair}}^k$ by replacing $\mathcal{D}_-$ with using $\mathcal{D}_-^k$ in Equation 12.
10:     **If** $b_{\mathcal{D}_{\text{val}}}(\hat{\boldsymbol{\theta}}_{\text{fair}}^k) < b_{\mathcal{D}_{\text{val}}}(\hat{\boldsymbol{\theta}}_{\text{fair}}^{k-1})$ **set** $\hat{\boldsymbol{\theta}}_{\text{fair}} := \hat{\boldsymbol{\theta}}_{\text{fair}}^k$ **else set** $\hat{\boldsymbol{\theta}}_{\text{fair}} := \hat{\boldsymbol{\theta}}_{\text{fair}}^{k-1}$ and break out of the for loop.
11: **end for**
12: **Return:** fair model parameters $\hat{\boldsymbol{\theta}}_{\text{fair}}$.

---

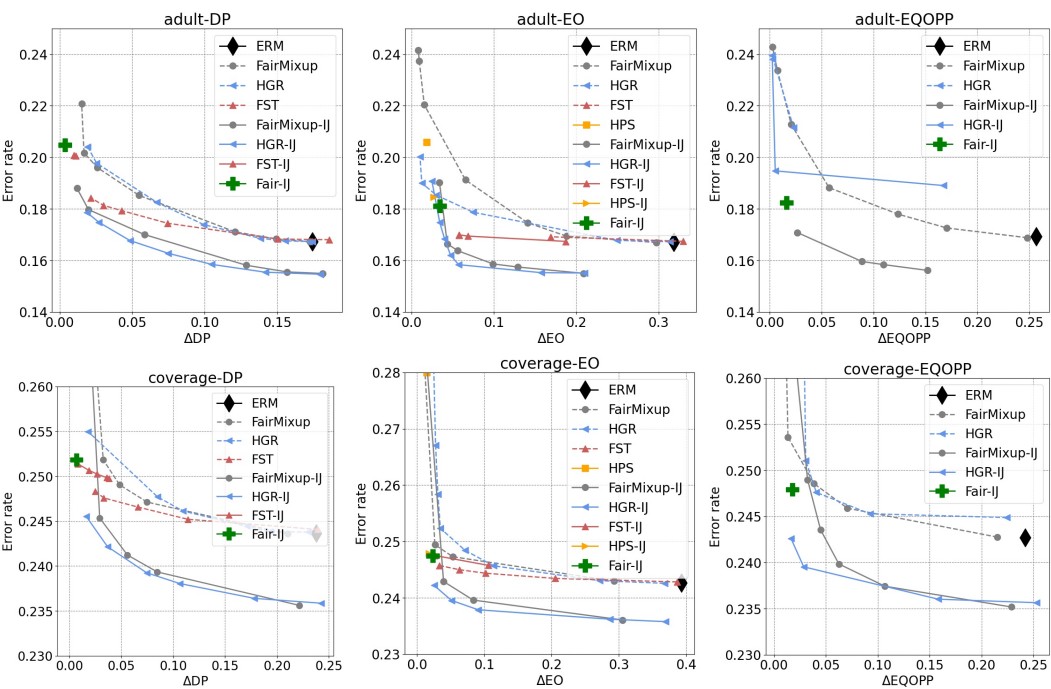

Figure 2: Accuracy and fairness (DP, EO, and EQOPP) Pareto frontier for the Adult and the Coverage datasets averaged over 10 runs. Points closer to the bottom-left achieve the best fairness/accuracy trade-off.

sensitive attribute. In the rest of the paper, we refer to this subset simply as *Coverage* dataset. We randomly split the dataset into train/validation/test with partition $50\% / 20\% / 30\%$, respectively, for each trial. Additionally, for both datasets we standardize the features before training our methods and the baselines.

We first train a 1-hidden layer fully connected neural network with *SeLu* activation function and 100 hidden units. This initial model is trained using standard ERM loss with batch size set to 256. We use the Adam optimizer with learning rate set to $10^{-4}$. We train the ERM and baselines for 100 epochs and pick the checkpoint with best accuracy on the validation set. We then employ Algorithm 1 to

arrive at the Fair-IJ solution. We select $k$ and the IHVP scaling term based on the same validation set used to compute the influence scores.

**Compared algorithms.** We compare our method to several in-processing and post-processing bias mitigation algorithms that are applicable to a wide range of model classes including deep neural networks. We omit the comparisons with pre-processing methods as our goal is to improve a given pre-trained model. Among in-processing algorithms, we compare with: *FairMixup* [9] and *HGR* [27]. *FairMixup* achieves fairness through a *mixup* based regularization employed during training. The *HGR* approach proposes a surrogate to HGR dependence measure and promotes fairness during training by enforcing conditional independence implied by the fairness metrics. On both of these methods, the fairness accuracy trade-off is achieved through the strength of the regularizer.

Among post-processing methods, we compare with *FST* [39] and *HPS* [17]. *FST* optimally transforms the pre-trained model's prediction scores to satisfy a specified fairness constraint and supports DP and EO metrics. To improve the performance of *FST* we re-calibrate the prediction scores using isotonic Regression. *HPS* is designed to enforce EO and requires knowledge of the sensitive attribute at test time. We use a fixed pre-trained model, architecture and training procedure for all the baselines. We also run the baselines on the edited model obtained from the output of our Fair-IJ algorithm, $\hat{\theta}_{\text{fair}}$. Specifically, we fine-tune the Fair-IJ solution using the in-processing algorithms. In the case of post-processing algorithms, we directly apply these to the Fair-IJ edited model.

**Results.** Figure 2 shows the accuracy and fairness Pareto frontier for the Adult and the Coverage datasets averaged over 10 runs. It can be seen that Fair-IJ consistently produces lower disparities across datasets and metrics. Moreover, we observe that baselines operating on $\hat{\theta}_{\text{fair}}$, FairMixup-IJ, HGR-IJ, FST-IJ, and HPS-IJ often achieve substantially better accuracy/fairness trade-off over their counterparts. In Figure 1, we plot the sorted influence scores of training instances on the average demographic parity of a held-out validation set for different datasets considered in this paper.

## 4.2 CivilComments Dataset

**Setup.** The *CivilComments* dataset [5] consists of human-annotated attributes on hate comments posted on the internet. The task here is to predict whether a particular comment is toxic. Prior work has shown that automatic toxicity classifiers can achieve sub-optimal performance on certain subpopulations [28, 12, 31]. The goal is to apply our approach to mitigate bias in pre-trained toxicity classifiers. In our experiments we consider Muslim as the sensitive attribute. Similar to [22], we assign an instance to the unprivileged group whenever it is annotated with that attribute and assign the rest to the privileged group.

To show the adaptability of our method on large neural networks, we consider three different variants of the pre-trained frozen BERT [10], where features are augmented with: a) BERT$_{\text{LC}}$: a linear classifier head, b) BERT$_{\text{NC}}$: a non-linear classifier head and c) BERT$_{\text{TT=n}}$: with $n$ trigger-tokens in the embedding layer. The last variant is an extension of prompt-tuning [23] or prefix-tuning [24] methods, which are more powerful ways of fine-tuning large-language models than only updating classifier heads. It is worth noting that the adaptation of trigger-tokens scale fittingly in optimizing weights in Equation 9. We compare our results to the simple yet effective method of Gap Regularization (GapReg) from [9] where a model optimization is regularized by a fairness measure added to the training loss while scaled by $\lambda$ factor to control the regularizer magnitude, as defined in Equation 1 of [9].

**Results.** We present our results in Table 1. In comparison to the baseline methods ERM and GapReg, Fair-IJ consistently performs better in mitigating the group disparities. Additionally, Fair-IJ manages to have a better task performance (balanced accuracy for toxicity classification) trade-off while attempting to achieve a lower disparity. In Table 1, we also present the results on virtual trigger-tokens, which we notice to be performing equally well in lowering the disparity. This is a significant observation as it shows how Fair-IJ can be efficiently integrated with large neural network through the scalable influence calculations of relatively few trigger parameters. Further training details, observations and baselines are presented in the Appendix.

## 5 Conclusion

In this work, we proposed Fair-IJ, an infinitesimal jackknife-based approach to mitigate the influence of biased training data points without refitting the model. Our approach is limited to

| Model | BA | ΔEO | BA | ΔDP | Model | BA | ΔEO | BA | ΔDP |
|---|---|---|---|---|---|---|---|---|---|
| BERT$_{LC}$-ERM | 57.3 | 0.314 | 58.4 | 0.246 | BERT$_{TT=4}$-ERM | 57.5 | 0.360 | 57.5 | 0.268 |
| BERT$_{LC}$-GapReg | 57.9 | 0.133 | 57.5 | 0.185 | BERT$_{TT=4}$-Fair-IJ | 56.0 | 0.102 | 55.2 | 0.042 |
| BERT$_{LC}$-Fair-IJ | 58.6 | 0.125 | 57.1 | 0.011 | BERT$_{TT=8}$-ERM | 58.2 | 0.317 | 58.2 | 0.254 |
| BERT$_{NC}$-ERM | 59.3 | 0.326 | 59.3 | 0.261 | BERT$_{TT=8}$-Fair-IJ | 56.5 | 0.113 | 56.4 | 0.071 |
| BERT$_{NC}$-GapReg | 59.1 | 0.144 | 58.2 | 0.054 | BERT$_{TT=10}$-ERM | 58.3 | 0.348 | 58.3 | 0.234 |
| BERT$_{NC}$-Fair-IJ | 59.8 | 0.126 | 58.6 | 0.008 | BERT$_{TT=10}$-Fair-IJ | 57.2 | 0.110 | 56.6 | 0.089 |

Table 1: Comparison between ERM, Gap Regularization (for $\lambda = 1$), and `Fair-IJ` for CivilComments on the sensitive attribute `MUSLIM` when we use pre-trained BERT model. We report the difference in equality of odds (ΔEO), difference in demographic parity (ΔDP), along with the task balanced accuracy (BA).

settings where the assumptions listed in Section 3 hold. Also, care must be taken to choose an appropriate fairness criterion and its differentiable surrogate for any given application to avoid unwanted consequences. We restricted our analysis to binary classification, since this is by far the most common setup in the fairness literature, but our approach applies to any metric $M$ that is once differentiable in the model parameters and any training loss that is twice differentiable in the parameters. This includes standard approaches to multiclass classification and regression.

Future work includes extending our approach to black-box models where the gradients are inaccessible and incorporating higher-order Taylor approximations to improve the accuracy of the influence functions. We hope that our method further encourages researchers and practitioners in studying and applying bias mitigation to diverse and complex models and datasets.

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
