# Appendix

## A  Loss aware $w_{\text{fair}}$

While this construction of $\mathbf{w}_{\text{fair}}$ reduces the fairness metric it pays no heed to the original loss, $\ell$, and may lead to classifiers that are less accurate than $\hat{\boldsymbol{\theta}}$. To account for $\ell$ we additionally compute $\mathcal{I}_{\ell,n}$ the influence of each training instance on the validation loss, $L_{\mathcal{D}_{\text{val}}}(\boldsymbol{\theta}) = \frac{1}{N_{\text{val}}} \sum_{n=1}^{N_{\text{val}}} \ell(y_n, h_{\boldsymbol{\theta}}(\mathbf{x}_n))$ and set,

$$w_n^{\text{fair}} = 1 - \mathbb{1}[\mathcal{I}_{M,n} > 0]\mathbb{1}[\mathcal{I}_{\ell,n} > 0], \tag{15}$$

i.e., we zero out those coordinates of $\mathbf{w}_{\text{fair}}$ that correspond to training instances with a positive influence on both the fairness metric and the loss $\ell$. Finally, defining $\mathcal{D}_- = \{\mathbf{z}_n \mid \mathbf{z}_n \in \mathcal{D} \text{ and } \mathcal{I}_{M,n} > 0 \ \& \ \mathcal{I}_{\ell,n} > 0\}$, we arrive at the post-hoc mitigated classifier by plugging in the zeroed-out $\mathbf{w}_{\text{fair}}$ in Equation 9,

$$\hat{\boldsymbol{\theta}}_{\text{fair}} = \hat{\boldsymbol{\theta}} + \sum_{m \in \mathcal{D}_-} H^{-1} g_m. \tag{16}$$

## B  IHVP-WoodFisher

In this section, we show that the coupled recurrences, that we refer to as *IHVP-WoodFisher*, computes the Inverse-Hessian Vector Product (IHVP). We begin by restating Proposition 3.2.

**Proposition B.1.** *Let $\boldsymbol{o}_1 = \nabla_\theta \ell(\mathbf{z}_1)$, $\boldsymbol{k}_1 = \mathbf{v}$, and $N$ denote the number of training instances. The Hessian-vector product $H^{-1}\mathbf{v}$ is approximated by iterating through the IHVP-WoodFisher recurrence in Equation 14 and computing $\boldsymbol{k}_N$.*

*Proof.* The WoodFisher approximation provides us with the following recurrence relation for estimating the inverse of the Hessian,

$$H_{n+1}^{-1} = H_n^{-1} - \frac{H_n^{-1} \nabla_\theta \ell(\mathbf{z}_{n+1}) \nabla_\theta \ell(\mathbf{z}_{n+1})^\top H_n^{-1}}{N + \nabla_\theta \ell(\mathbf{z}_{n+1})^\top H_n^{-1} \nabla_\theta \ell(\mathbf{z}_{n+1})} \tag{17}$$

with, $H_0^{-1} = \lambda^{-1} I_D$, and $\lambda$ is small positive scalar.

Multiplying, both sides by $\nabla_\theta \ell(\mathbf{z}_{n+1})$, we get,

$$H_{n+1}^{-1} \nabla_\theta \ell(\mathbf{z}_{n+1}) = H_n^{-1} \nabla_\theta \ell(\mathbf{z}_{n+1}) - \frac{H_n^{-1} \nabla_\theta \ell(\mathbf{z}_{n+1}) \nabla_\theta \ell(\mathbf{z}_{n+1})^\top H_n^{-1} \nabla_\theta \ell(\mathbf{z}_{n+1})}{N + \nabla_\theta \ell(\mathbf{z}_{n+1})^\top H_n^{-1} \nabla_\theta \ell(\mathbf{z}_{n+1})} \tag{18}$$

By substituting $H_n^{-1} \nabla_\theta \ell(\mathbf{z}_{n+1})$ with $\boldsymbol{o}_n$ and assuming $\nabla_\theta \ell(\mathbf{z}_{n+1})$ and $\nabla_\theta \ell(\mathbf{z}_{n+2})$ are close, we construct the following recurrence relation

$$\boldsymbol{o}_{n+1} = \boldsymbol{o}_n - \frac{\boldsymbol{o}_n \nabla_\theta \ell(\mathbf{z}_{n+1})^\top \boldsymbol{o}_n}{N + \nabla_\theta \ell(\mathbf{z}_{n+1})^\top \boldsymbol{o}_n} \tag{19}$$

Now, multiplying both sides of Equation 17 by $\mathbf{v}$, gives us the recurrence relation for the IHVP,

$$H_{n+1}^{-1} \mathbf{v} = H_n^{-1} \mathbf{v} - \frac{H_n^{-1} \nabla_\theta \ell(\mathbf{z}_{n+1}) \nabla_\theta \ell(\mathbf{z}_{n+1})^\top H_n^{-1} \mathbf{v}}{N + \nabla_\theta \ell(\mathbf{z}_{n+1})^\top H_n^{-1} \nabla_\theta \ell(\mathbf{z}_{n+1})} \tag{20}$$

By substituting $H_{n+1}^{-1} \mathbf{v}$ with $\boldsymbol{k}_{n+1}$ and $H_n^{-1} \nabla_\theta \ell(\mathbf{z}_{n+1})$ with $\boldsymbol{o}_n$, we get

$$\boldsymbol{k}_{n+1} = \boldsymbol{k}_n - \frac{\boldsymbol{o}_n \nabla_\theta \ell(\mathbf{z}_{n+1})^\top \boldsymbol{k}_n}{N + \nabla_\theta \ell(\mathbf{z}_{n+1})^\top \boldsymbol{o}_n} \tag{21}$$

Thus, under our assumptions, when $H_n^{-1}$ converges to $H^{-1}$, $\boldsymbol{k}_n$ converges to the IHVP $H^{-1}\mathbf{v}$.

$\square$

## C   IHVP-WoodFisher Approximation Accuracy

In this section, we discuss the approximation accuracy of *IHVP-WoodFisher* and compare it with *IHVP-Neumann*.

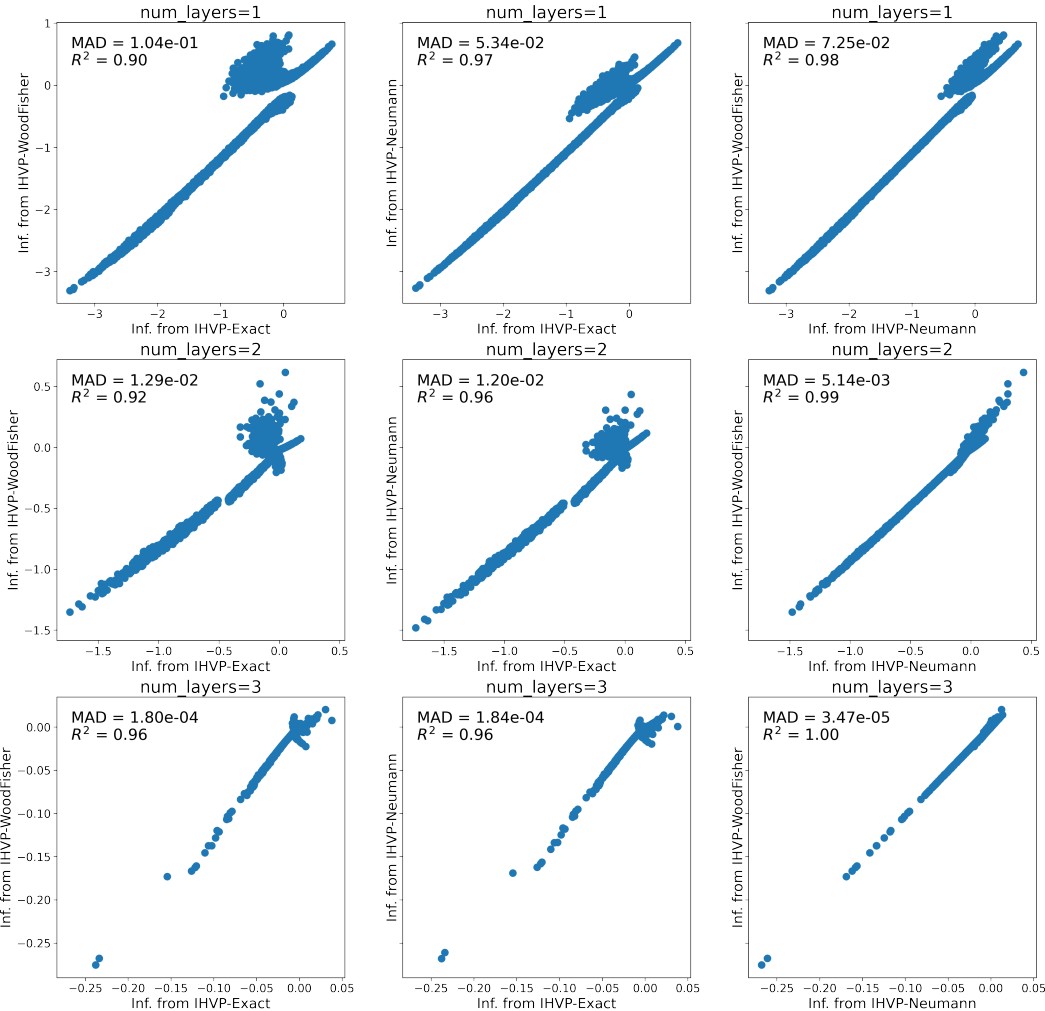

Figure 3: **IHVP-WoodFisher Approximation Accuracy.** The plots show the accuracy of the *IHVP-WoodFisher* and *IHVP-Neumann* approximations by comparing with influence scores obtained from *IHVP-Exact*. We train models with different number of layers on the linearly separable two moon dataset and report Median Absolute Deviation (MAD) and $R^2$ scores.

*IHVP-WoodFisher* relies on the assumption that the empirical Fisher matrix is a good approximation to the Hessian of the loss. The Hessian of the loss is known to converge to the true Fisher matrix, when: a) the loss of the model used during training can be expressed as negative log likelihood; and b) the model likelihood has converged to the true data likelihood [35]. The empirical Fisher matrix does not have convergence guarantees as the true Fisher but, it is computationally cheap and works well in practice as an approximation to the Hessian matrix. This is also seen in our experimental results.

To study the approximation accuracy of *IHVP-WoodFisher* and compare with the *Exact* approach of computing IHVP, we consider a setting where the number of parameters is small; specifically, we generate a linearly separable variant of the two moon dataset consisting of $10000$ points, where each point has 2 input features and can belong to one of the two classes. We create at $80 - 20$ train-test split and train models with depth 1, 2, and 3 to observe the effect of depth. Hidden layers have a fixed width of 5 units. We use Adam optimizer with learning rate $0.001$. After training, we pick

**Algorithm 2** `Fair-IJ (slow)`

---

1: **Input:** Pre-trained model parameters $\hat{\boldsymbol{\theta}}$, training set $\mathcal{D}$, loss function $\ell$, a validation set $\mathcal{D}_{\text{val}}$ and a smooth surrogate to the fairness metric $b \in \{\Delta\text{DP}, \Delta\text{EO}\}$, $M^b_{\mathcal{D}_{\text{val}}}$.

2: **Calculate:** $\mathbf{H}^{-1} g_n$ for each training instance $\mathbf{z}_n$ by setting $\boldsymbol{k}_1 = g_n$ and iterating through Equation 14 for B iterations.

3: **Calculate:** the fairness influence $\mathcal{I}_{b,n}$ of each training instance $\mathbf{z}_n$ on $\mathcal{D}_{\text{val}}$ using Equation 7 or Equation 8.

4: **Construct:** the set $\mathcal{D}_-$ and denote its cardinality, $|\mathcal{D}_-| = K$.

5: **Initialize:** $\hat{\boldsymbol{\theta}}^0_{\text{fair}} := \hat{\boldsymbol{\theta}}$

6: **for** $k \in [1, \ldots, K]$ **do**

7:      **Construct:** $\mathcal{D}^k_- = \{\mathbf{z}_n \in \mathcal{D}_- \mid \mathcal{I}_{b,n} > \mathcal{I}_{b,(K-k)}\}$, where $\mathcal{I}_{b,(K-k)}$ denotes the $(K-k)^{\text{th}}$ order statistic of the influence scores $[\mathcal{I}_{b,1}, \ldots, \mathcal{I}_{b,K}]$.

8:      **Calculate:** $\hat{\boldsymbol{\theta}}^k_{\text{fair}}$ by replacing $\mathcal{D}_-$ with using $\mathcal{D}^k_-$ in Equation 12.

9:      **If** $b_{\mathcal{D}_{\text{val}}}(\hat{\boldsymbol{\theta}}^k_{\text{fair}}) < b_{\mathcal{D}_{\text{val}}}(\hat{\boldsymbol{\theta}}^{k-1}_{\text{fair}})$ **set** $\hat{\boldsymbol{\theta}}_{\text{fair}} := \hat{\boldsymbol{\theta}}^k_{\text{fair}}$ **else set** $\hat{\boldsymbol{\theta}}_{\text{fair}} := \hat{\boldsymbol{\theta}}^{k-1}_{\text{fair}}$ and break out of the for loop.

10: **end for**

11: **Return:** fair model parameters $\hat{\boldsymbol{\theta}}_{\text{fair}}$.

---

a random point from the test set and compute the influence score of the training instances using *IHVP-WoodFisher* and *IHVP-Neumann* approximations as well as exactly computing the IHVPs. For both approximations we use 1000 iterations and average over 10 runs. The *IHVP-Neumann* approximation has an additional hyper-parameter - *scale*. This is to ensure that the Eigenvalues of the Hessian are between $[0, 1]$. For *IHVP-Neumann*'s convergence, *scale* has to be greater than the largest Eigenvalue of the Hessian. In these experiments, we set this hyperparameter to $25.0$ which is larger than the largest Eigenvalue we observed for all the models we trained.

In Figure 3 we compare the influence scores and report the Median Absolute Deviation (MAD) and $R^2$ scores. For each depth value, when plotting and computing the metrics we rescale the influence scores from both approximations to match the mean of the *IHVP-Exact*. It can be observed that both *IHVP-Neumann* and *IHVP-WoodFisher* approaches match well with the influence scores obtained from *Exact* computation. The main benefit of *IHVP-WoodFisher* is that it does not require higher-order gradients. Additionally, unlike *IHVP-Neumann*, this method does not require expensive hyperparameter search to rescale the Eigenvalues of Hessian to ensure IHVP computation convergence.

# D    Additional Dataset, Training Details and Results

In Table 2, we provide additional information information about the three datasets – Adult[2], Coverage [3], and CivilComments[4]. We trained our models on NVIDIA A100 Tensor Core GPUs. In the case of the tabular datasets, 10 runs with a particular fairness metric took less than 2 hours. This includes training the base model using ERM followed by the application of our `Fair-IJ` algorithm. Within the algorithm, we search for the best $k$ among $40$ values spread uniformly in the range in the ranges $0 - 2000$. Similarly, for the IHVP scaling we select the best value among $(0.01, 0.1, 1.0, 2.0, 3.0, 5.0, 10.0)$. This search only requires inference over the validation and hence is relatively inexpensive. The post-processing baselines (*FST* and *HPS*), assume access to a pre-trained model. Similar to our approach they use the validation data to mitigate bias in the pre-trained models. For the in-processing baselines (*HGR* and *FairMixup*), following standard practice, we train the models on the training set and use the validation set to select the hyper-parameter that determines the strength of the fairness regularizer employed by these methods. In Figure 4, we reproduce the Figure 2 with error bars for both the accuracy and fairness metrics.

We now provide additional details regarding the experiments on CivilComments datasets. Table 3 presents additional results comparing ERM models (built without any fairness adjustment) to models regularized using Gap Regularization (GapReg) from *FairMixup* [9], Hirschfeld-Gebelein-Rényi

---

[2] https://archive.ics.uci.edu/ml/datasets/adult
[3] https://github.com/zykls/folktables
[4] https://www.kaggle.com/c/jigsaw-unintended-bias-in-toxicity-classification/data

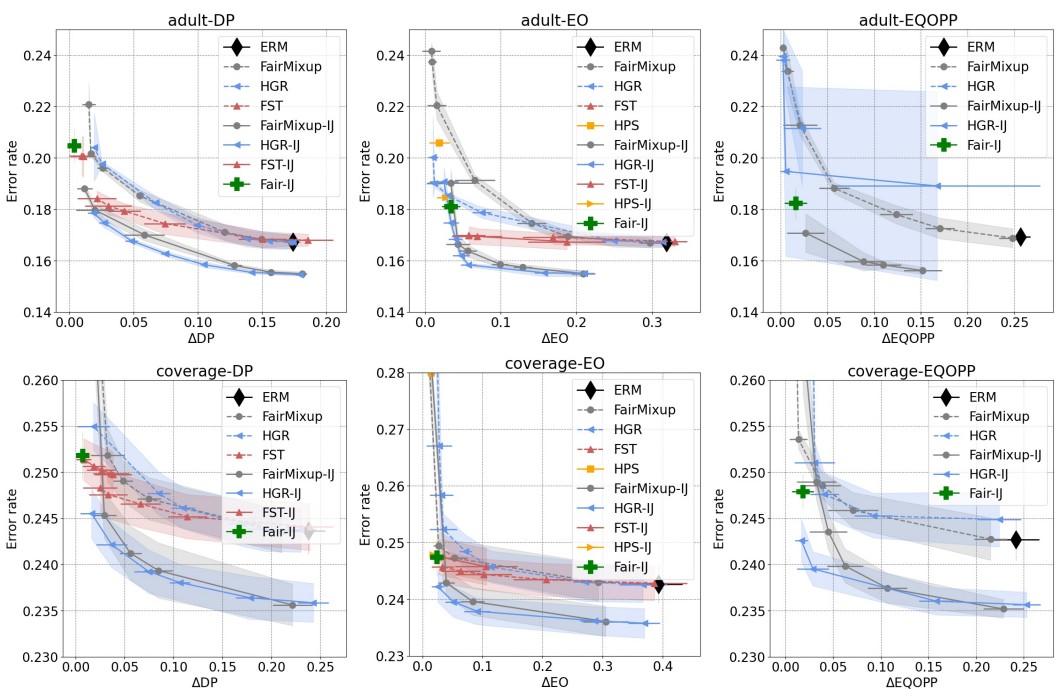

Figure 4: Accuracy and fairness Pareto frontier with error bars for the Adult and the Coverage datasets averaged over 10 runs. Points closer to the bottom-left achieve the best fairness/accuracy trade-off.

Table 2: Summary of datasets.

| Dataset | Target | Attribute | Size |
| :---: | :---: | :---: | :---: |
| | | | Train / Valid / Test |
| Adult | Income | Sex | 21815 / 10746 / 12661 |
| Coverage | Health Insurance Coverage | Race | 44168 / 21755 / 32471 |
| Civil Comments | Toxicity | Muslim | 269038 / 45180 / 133782 |

Maximum Correlation Coefficient (HGR) dependency measure [27], and *Fair-IJ*. Reported results are for the sensitive attribute MUSLIM. They include Equality of odds ($\Delta$EO), demographic parity ($\Delta$DP), and balanced accuracy (BA) and are the means and standard deviations over 5 training runs, each with a different random seed (i.e. 5 different seeds for each configuration). Each model was built with 100 epochs of SGD over the training data for a total of 24h of computation time (using a NVIDIA A100 GPU). All reported results were computed on the test dataset for models with the best validation loss over the 100 epochs of training (models being validated at the end of each epoch). For loss-regularized models using GapReg and HGR, values of $\lambda = 1.0$ and $\lambda = 0.2$ were used to add the regularizer term to the training loss; both sets of results are given in Table 3. Inference on the test dataset is quite fast (3 minutes) on NVIDIA A100 GPUs. Results for both BERT$_{LC}$ (Linear Classifier) and BERT$_{NC}$ (Non-linear Classifier) are provided for ERM, GapReg, HGR, and `Fair-IJ`.

Both our BERT$_{LC}$ and BERT$_{NC}$ architectures are defined based on different classification layer(s) on top of a pre-trained BERT model. In all our experiments, we use BERT$_{base}$ model. While BERT$_{LC}$ uses just a single dense layer on top of the pooled vector from BERT representations, BERT$_{NC}$ has multiple dense layers on top of the pooled output. For the latter, we use two dense layers with hidden sizes of 768 and 128 intertwined with ReLU non-linearities. As described in 4.2, we also provide quantitative results on a variant of BERT$_{LC}$, referred as BERT$_{TT=N}$, which uses virtual tokens, where N refers to number of trigger tokens. In this setup, we introduce a new parameterized embedding, called *trigger* embeddings, which are learned during the training. Similar to methods introduced in [23] and [24], we add trigger tokens to each sequence during the training. In our quantitative analysis we use

| Model | BA | $\Delta$EO | BA | $\Delta$DP |
|---|---|---|---|---|
| BERT$_{\text{LC}}$-ERM | 58.4±0.229 | 0.314±0.026 | 58.4±0.229 | 0.246±0.011 |
| BERT$_{\text{NC}}$-ERM | 59.3±0.025 | 0.326±0.006 | 59.3±0.025 | 0.261±0.003 |
| BERT$_{\text{LC}}$-GapReg ($\lambda=0.2$) | 58.3±0.000 | 0.198±0.006 | 58.3±0.000 | 0.149±0.004 |
| BERT$_{\text{LC}}$-GapReg ($\lambda=1.0$) | 57.9±0.001 | 0.144±0.010 | 56.9±0.016 | 0.071±0.009 |
| BERT$_{\text{NC}}$-GapReg ($\lambda=0.2$) | 59.0±0.000 | 0.166±0.007 | 58.8±0.005 | 0.136±0.010 |
| BERT$_{\text{NC}}$-GapReg ($\lambda=1.0$) | 57.9±0.011 | 0.144±0.067 | 58.2±0.003 | 0.054±0.028 |
| BERT$_{\text{LC}}$-HGR ($\lambda=0.2$) | 56.9±0.018 | 0.209±0.075 | 58.3±0.000 | 0.213±0.002 |
| BERT$_{\text{LC}}$-HGR ($\lambda=1.0$) | 53.0±0.002 | 0.372±0.015 | 58.1±0.000 | 0.149±0.003 |
| BERT$_{\text{NC}}$-HGR ($\lambda=0.2$) | 59.0±0.000 | 0.169±0.007 | 59.1±0.003 | 0.240±0.004 |
| BERT$_{\text{NC}}$-HGR ($\lambda=1.0$) | 54.0±0.003 | 0.339±0.042 | 59.1±0.000 | 0.191±0.002 |
| BERT$_{\text{LC}}$-Fair-IJ | 58.6±0.162 | 0.125±0.011 | 57.1±0.237 | 0.011±0.004 |
| BERT$_{\text{NC}}$-Fair-IJ | 59.8±0.326 | 0.126±0.011 | 58.6±0.170 | 0.008±0.004 |

| Model | BA | $\Delta$EO | BA | $\Delta$DP |
|---|---|---|---|---|
| BERT$_{\text{TT=4}}$-ERM | 57.5±0.794 | 0.360±0.090 | 57.5±0.794 | 0.268±0.037 |
| BERT$_{\text{TT=8}}$-ERM | 58.2±0.179 | 0.317±0.060 | 58.2±0.179 | 0.254±0.032 |
| BERT$_{\text{TT=10}}$-ERM | 58.3±0.842 | 0.348±0.070 | 58.3±0.842 | 0.234±0.043 |
| BERT$_{\text{TT=4}}$-Fair-IJ | 56.0±0.922 | 0.102±0.035 | 55.2±1.202 | 0.042±0.060 |
| BERT$_{\text{TT=8}}$-Fair-IJ | 56.5±0.690 | 0.113±0.040 | 56.4±0.512 | 0.071±0.069 |
| BERT$_{\text{TT=10}}$-Fair-IJ | 57.2±1.483 | 0.110±0.045 | 56.6±1.349 | 0.089±0.056 |

Table 3: Comparison between ERM, Gap Regularization, Hirschfeld-Gebelein-Renyi Maximum Correlation Coefficient (HGR) (both for $\lambda = 0.2, 1.0$), and Fair-IJ for CivilComments on the sensitive attribute MUSLIM. We report the mean and standard deviation of difference in equality of odds ($\Delta$EO), difference in demographic parity ($\Delta$DP), along with the task balanced accuracy (BA) on 5 different seeds.

variants with 4, 8, and 10 trigger tokens which are referred as BERT$_{\text{TT=4}}$, BERT$_{\text{TT=8}}$, and BERT$_{\text{TT=10}}$ respectively in Table 3 and Table 1. We fine-tune these models with a maximum epochs of 100 and choose the best model based on the validation loss over the validation set. Similar to the case of tabular datasets, we apply Fair-IJ algorithm with same range of $k$ and IHVP scaling.

| Model | BA | $\Delta$EO | BA | $\Delta$DP |
|---|---|---|---|---|
| T5$_{\text{TT=4}}$-ERM | 59.9 | 0.150 | 59.9 | 0.170 |
| T5$_{\text{TT=4}}$-Fair-IJ | 52.8 | 0.019 | 55.2 | 0.008 |
| T5$_{\text{TT=8}}$-ERM | 59.1 | 0.141 | 59.1 | 0.157 |
| T5$_{\text{TT=8}}$-Fair-IJ | 52.6 | 0.019 | 56.4 | 0.002 |
| T5$_{\text{TT=10}}$-ERM | 59.3 | 0.150 | 59.3 | 0.158 |
| T5$_{\text{TT=10}}$-Fair-IJ | 54.5 | 0.027 | 54.9 | 0.004 |

Table 4: Comparison between ERM and Fair-IJ for CivilComments on the sensitive attribute MUSLIM when we use pre-trained T5 model. We report the difference in equality of odds ($\Delta$EO), difference in demographic parity ($\Delta$DP), along with the task balanced accuracy (BA).

Further to validate our approach on larger models, we performed additional experiments (Table 4) with a much larger transformer model T5, on the CivilComments dataset with the sensitive attribute set to "Muslim" (i.e., the same setup as 4.2). Our results show similar trends to the experiments with BERT. Fair-IJ consistently decreases the fairness disparity between groups over the empirical risk minimization solution.