# OpenReview forum: "Fair Infinitesimal Jackknife: Mitigating the Influence of Biased Training Data Points Without Refitting"
_NeurIPS.cc/2022/Conference — NeurIPS 2022 Accept_

### Official Review · Reviewer_T4TM · 2022-07-06

**Rating:** 5
**Confidence:** 4
**Soundness:** 3 good
**Presentation:** 2 fair
**Contribution:** 3 good

**Summary:**

The draft proposes to simulate removal, i.e. without actually retraining the model, of training points highly influential on a fairness surrogate metric. The work builds upon the concept of influence functions and reuses them beyond loss functions to estimate the effect on other functions of parameters. The paper also presents a method to avoid the O(d^2) complexity of running the Wood-Fisher inverse Hessian approximation procedure (however there seem to exist better alternatives, see below). The results improve on top of the existing baselines by achieving smaller violations of the fairness constraints, while not much reducing the base task accuracy.



**Questions:**

The draft lacks a reference to a faster alternative to Wood-Fisher method for efficient influence functions calculation and to increase their accuracy: https://arxiv.org/abs/2112.03052. They compute an approximate Hessian, which is storable in the basis of few dozens of largest eigenvalues of the true Hessian, so that getting $H^{-1}$ is extremely cheap in that basis. With this, there would be no need to rerun iterations of Wood-Fisher for every point $v$, and influence calculation boils down to a few matrix multiplications.

Could you explain in more detail the modifications to BERT? Do BERT_LC and BERT_NC sit on top of the CLS token embeddings?

Minor:
- inconsistent use of FairIJ vs Fair-IJ in text and in figures, and also in different fonts
- 206: Does 'IJ' stand for infinitesimal jackknife? Please introduce it explicitly.
- 227: Please state preferably earlier that you dubbed your approach FairIJ (the algorithm caption is easy to miss)


**Limitations:**

It's unclear how this work would scale to truly large models given the need of recomputing the iterative procedure for every validation example and over all training points.


**Strengths And Weaknesses:**

Strengths:
- a nice repurposing of the influence function approach to other functions than just the loss
- strong results in reducing disparity without hurting the accuracy too much

Weakness:
- Potentially slow influence calculations (see the Questions section)
- While the draft seem to use BERT as a stand-in for large (language) models, in reality it's very small compared to them.
- Writing could be improved by carefully introducing notation and abbreviations, and spending more time on model descriptions (e.g. BERT)

---

> ### Author Response · Authors · 2022-08-02
> **Response**
>
> **We thank the reviewer for the constructive comments. Please find our responses below addressing reviewers’ specific concerns and questions.**
>
> - **Potentially slow influence calculation** -- This is a great point; thank you for bringing this up. Although Algorithm 1 suggests running the Wood-Fisher iterations for each training instance for clarity of exposition, in practice (see lines 169-199 in the file *influence_utils.py* of our submitted code), we use the following trick to run the Wood-Fisher iterations only once for the entire training dataset. First, for any $p$ \* $p$ symmetric matrix $\mathbf{A}$ and $p$-dimensional vectors $\mathbf{x}$ and $\mathbf{y}$,  $\mathbf{x}^T \mathbf{A} \mathbf{y}$ = $\mathbf{y}^T \mathbf{A} \mathbf{x}$. From Equation 5, the influence calculation involves computing $\nabla_\theta M(\theta(\mathbf{w}), \mathbf{w})^T H^{-1} g_n$ for all $n$ in the training dataset. Since $H$ is symmetric, we can equivalently compute $g_n^T H^{-1}\nabla_\theta M(\theta(\mathbf{w}), \mathbf{w})$. We can then run the Wood-Fisher iterations to approximate $H^{-1}\nabla_\theta M(\theta(\mathbf{w}), \mathbf{w})$. Crucially, we need to do this only once. With the approximation in hand, computing the per data influence requires a single dot product per data instance between $g_n$ and the Wood-Fisher approximated $H^{-1}\nabla_\theta M(\theta(\mathbf{w}), \mathbf{w})$.   Other approaches to speeding up and/or improving the accuracy of the influence estimation, including the very nice work you point out, could certainly be used in our framework. While the Wood-Fisher approximation as used above requires us to store a single $D$-dimensional vector, the work in https://arxiv.org/abs/2112.03052 would require the storage of a $D$\*$K$ matrix (for $K$ largest eigenvectors) and a matrix multiplication instead of a vector dot product per training point. We will add a section discussing these computational tradeoffs to the manuscript and cite the new work you point out.
>
> - **Could you explain in more detail the modifications to BERT? Do BERT_LC and BERT_NC sit on top of the CLS token embeddings?**  --  BERT_LC and BERT_NC correspond to two variants of BERT that use different classifier heads. The classifier head sits on top of the CLS token embedding.  BERT_LC (Linear classifier) uses a single dense layer, while BERT_NC (Non-Linear classifier) uses multiple dense layers. The specific architectural details are documented in the appendix (Section D, lines 512 -- 524). We will additionally update the experimental section of the main paper to clarify this.
>
> - **BERT is small** -- We demonstrated the efficacy of our approach for a transformer-based model, BERT. Our goal with this experiment was to merely demonstrate the compatibility of our approach with this class of models. Nonetheless, we agree that by modern standards, Bert is small. To further validate our approach on larger models, we performed additional experiments with a much larger model --- T5, on the CivilComments dataset with the sensitive attribute set to “Muslim” (i.e., the same setup as section 4.2). Our preliminary results show similar trends to the experiments with BERT. FairIJ consistently decreases the fairness disparity between groups over the empirical risk minimization solution. We will include a detailed discussion of the T5 results in the final version, should the paper be accepted.
>
> | Model      | BA | $\Delta\text{EO}$ | BA | $\Delta\text{DP}$ |
> | ----------- | ----------- | ----------- | ----------- |----------- |
> | ${T5_{TT=4}}-\texttt{ERM}$        | 0.593913        | 0.191049         | 0.593913        | 0.187675       |
> | ${T5_{TT=8}}-\texttt{ERM}$  | 0.594919        | 0.205861        | 0.594919        |0.198395        |
> | ${T5_{TT=10}}-\texttt{ERM}$   | 0.592045        | 0.215676        | 0.592045        |0.176334        |
> | ${T5_{TT=4}}-\texttt{FairIJ}$   | 0.729753        | 0.007000        | 0.690469        |0.001800        |
> | ${T5_{TT=8}}-\texttt{FairIJ}$   | 0.652768        | 0.014839        | 0.632169        |0.016691        |
> | ${T5_{TT=10}}-\texttt{FairIJ}$   | 0.629456        | 0.021317        | 0.614590        |0.013042        |
>
> - **Typos** – Thank you for catching these typos. We will fix them.

---

### Official Review · Reviewer_Xt7c · 2022-07-08

**Rating:** 6
**Confidence:** 4
**Soundness:** 3 good
**Presentation:** 2 fair
**Contribution:** 3 good

**Summary:**

The paper proposes a post-processing debiasing algorithm for fairness using an infinitesimal jackknife approach. Their approach examines the influence function of a model's parameters with respect to a fairness metric (demographic parity / equality of odds) to provide an update on the parameters. This comes in the form of a weighted gradient update (newton step), where the weights are chosen to improve fairness. This update requires the computation of an inverse Hessian vector calculation, where the paper proposes the use of an iterative calculation based on the WoodFisher approximation of Hessians. Experiments are also conducted, in which their algorithm is shown to improve fairness at the cost of some accuracy. Experiments also show that the infinitesimal jackknife can be used in conjunction with other in-processing / post-processing methods to potentially improve fairness and accuracy trade-off.

A few additional notes about the paper's approach:
  - Not only is the training data required, access to the parameters of the pre-trained model is needed
  - For Proposition 3.1 to allow for a reduction in the fairness penalty, the first order approximation of the criteria needs to be accurate at $ \mathbf{w}_{\rm fair} $
  - Only $k$ coordinate updates (wrt training data) are used in practice and not all given by Proposition 3.1.

**Questions:**

Questions / Comments / Suggestions
1. Although (from the narrative) it seems that training data is typically used to calculate the influence function and updates, what happens if one only updates the parameters with a validation dataset? Perhaps this would not be problematic if the pre-trained model / algorithm has strong generalization guarantees?
2. On Line 199, the trade-off between zeroing high influence coordinates and "accuracy / approximation quality". In experiments, the "best $k$" (and "IVHP scaling") is selected. What is "best" here? (validation loss or fairness?) Furthermore, what does the trade-off look like here (between $k$ and wrt the definition of "best")?
3. For the baseline approaches in the experiments, what data is given to these algorithms? Since the IJ update requires the training data + validation data, are both of these (concatenated?) passed to the baseline approaches?
4. Do the baseline post-processing approaches require access to the pre-trained model parameters (/ training data)? If I recall correctly RST and HPS do not. It would be useful for clarification on whether or not the requirements of the IJ algorithm are shared amongst the baselines.

**Limitations:**

The limitations / requirements of the paper's proposed approach are documented clearly. However, I think that comments on whether or not the compared baselines share these requirements should be made explicitly. (Reiterated from above)

---

Edit: The limitations were regarding the requirements of baselines was answered in the authors' response.

**Strengths And Weaknesses:**

Strengths
  - Experimentally the IJ algorithm has shows cases of strong performance. Particularly in combination with baseline fairness algorithms, which in some settings seem to be a strict improvement from running the baseline fairness algorithms alone.
  - The update on the parameters is intuitive, as described before Proposition 3.1.

Weaknesses
  - The access to the training data (and the model parameters) would not be applicable to all scenarios.
  - Some aspects of the experimental section are not clear, ie, training of baselines.

---

> ### Author Response · Authors · 2022-08-02
> **Response**
>
> **We thank the reviewer for their helpful feedback. Please find our responses below addressing reviewer’s questions.**
>
> - **Access to the training data** – The influence computation does need access to the data used to fit \theta. Given a large model pre-trained on a massive dataset and finetuned on a smaller task specific dataset, our approach only requires access to the smaller dataset used for finetuning the model (fitting $\mathbf{\theta}$), not the larger pre-training dataset. It is not uncommon for a user to have access to such finetuning data, even if the larger pre-training data may be unavailable to them. Indeed this is the setup we consider in the CivilComments experiment described in Section 4.2 of the paper. Of course, if even the finetuning data is unavailable then our approach cannot be used.
>
> - **Although (from the narrative) it seems that training data is typically used to calculate the influence function and updates, what happens if one only updates the parameters with a validation dataset? Perhaps this would not be problematic if the pre-trained model / algorithm has strong generalization guarantees?** --- As indicated above, if one uses a separate validation set to update the model (i.e., fit \$\mathbf{\theta}$), then our approach requires that the validation data be used to compute the influence on fairness. In terms of performance, how well this would work depends on both the size of the validation set --- the quality of the infinitesimal jackknife approximation improves with increasing data and the similarity between the validation and the test data distributions.
>
> - **On Line 199, the trade-off between zeroing high influence coordinates and "accuracy / approximation quality". In experiments, the "best k" (and "IVHP scaling") is selected. What is "best" here? (validation loss or fairness?) Furthermore, what does the trade-off look like here (between k and wrt the definition of "best")?** --- We select K and the IHVP scaling parameter based on the fairness metric. The selected K and IVHP scaling parameter correspond to values that achieve the lowest fairness disparity on the validation set. We observe that, initially, increasing K results in substantial drop in fairness metric with minimal drop in accuracy on the validation and the test set.  These instances are the ones with the highest fairness influence scores. Once K is large enough to include instances with lower fairness influence scores, we find that the fairness disparity decreases are more modest and on occasions can also marginally increase due to Taylor expansion errors (recall that we perform a Taylor expansion about the vector of all ones, and the error in the approximation grows as we evaluate at vectors far from the vector of all ones). We will clarify this in the paper.
>
> - **For the baseline approaches in the experiments, what data is given to these algorithms? Since the IJ update requires the training data + validation data, are both of these (concatenated?) passed to the baseline approaches?** --- The post-processing baselines (FST and HPS), assume access to a pre-trained model. Similar to our approach they use the validation data to mitigate bias in the pre-trained models.  For the in-processing baselines (HGR and FairMixup), following standard practice, we train the models on the training set and use the validation set to select the hyper-parameter that determines the strength of the fairness regularizer employed by these methods.
>
> - **Do the baseline post-processing approaches require access to the pre-trained model parameters (/ training data)? If I recall correctly RST and HPS do not. It would be useful for clarification on whether or not the requirements of the IJ algorithm are shared amongst the baselines.**  ---  The post-processing methods we compared against do not require access to the training data or the model parameters. Our method does require access to the training data. In exchange for this additional requirement, we find that our approach typically substantially outperforms these post-processing techniques. Crucially, similar to these post-processing approaches, our method mitigates bias in pre-trained models without requiring any additional refitting of the models. We did briefly mention these differences in lines 33-34 of the paper and will additionally highlight the training details of various baselines in the experimental section to clarify these issues further.

---

> > ### Comment · Reviewer_Xt7c · 2022-08-08
> > **Thank you for the response**
> >
> > Thanks you for the response.
> > I am happy with the response and appreciate the further clarification for some of the explanations I have missed. I believe that discussion and clarification in your response would help with the clarity of the paper. Especially the additional clarification referring to the requirements of the different baselines compared against (in particular the requirements of knowing the model parameters).
> >
> > I will wait for discussion with other reviewers before changing the numeric score. Nevertheless, I will edit my review lightly to indicate that the limitations have been answered adequately in your response.

---

### Official Review · Reviewer_38zQ · 2022-07-19

**Rating:** 6
**Confidence:** 3
**Soundness:** 3 good
**Presentation:** 3 good
**Contribution:** 3 good

**Summary:**

The paper proposes an infinitesimal jackknife-based approach to mitigate the influence of biased training data points without refitting the model. At a high level, the approach firstly computes the “influence” of training data points on the group fairness metrics of interest (e.g., demographic parity and equality of opportunity considered in this paper) using the coordinate-wise gradient of the first order Taylor approximation to the weighted risk-minimized model parameter, then drop training data points that have a disproportional impact on group (un)fairness. The paper provides theoretical analysis and empirical experiments for the performance of the proposed approach.

**Questions:**

Line 66, I may miss something, to compute the empirical risk minimization with dropping certain data points, the denominator in eq 2 should be the number of remaining data points?

Line 152, the authors say that the weights for each data instance are restricted to be binary. Any motivations behind this restriction?

Line 2 in Algorithm 1: what does B stand for?

Line 3 in Algorithm 1: the algorithm requires computing the fairness influence of each training data. But the definition of the fairness influence is from the true underlying data distribution and is computed in an expected manner. How do the authors compute these quantities given only access to the training data?


Other comments:
Usually, you can write the notations outside of your theorem/proposition environment so that the theorem/proposition statements can be more succinct.



**Limitations:**

See above

**Strengths And Weaknesses:**

The problem -- mitigating biases in machine learning models --  studied in this paper is definitely well-motivated. The proposed approach is (to my best knowledge) novel to me. The approach proposed in the paper that “mitigates pre-trained models without requiring any additional refitting of the model”, I think, should be useful when the pre-trained model is large and incur too much computation cost to train the model again. The authors also provide light theoretical analysis on why their proposed approach might work and also comprehensive experiments to validate the approach.

The proposed approach seems to only work for the binary classifier (at least the theoretical analysis is limited and only serves for binary classification). Adding results for multi-label classification to this paper would make it stronger. The theoretical result only establishes conditions under which the proposed approach could lead to classifiers with lower group disparities, it is hard to tell, from the experiment perspective, whether the proposed approach has a uniform higher accuracy compared with other approaches. It may greatly improve the paper if there exist theoretical guarantees for the achieved accuracy of the proposed method.

---

> ### Author Response · Authors · 2022-08-02
> **Response**
>
> **We thank the reviewer for their thoughtful comments. We address specific concerns raised in the reviews below.**
>
>
> - **Limited to Binary Classification** – Our approach (and analysis) applies to any function M that is once differentiable in the parameters \theta and any training loss that is twice differentiable in the parameters \theta. This includes standard approaches to multiclass classification and regression. In the paper, we focus on binary classification since this is by far the most common setup in the fairness literature. Although different ways of generalizing binary fair classification to regression and multiclass classification problems have been proposed, the community is yet to settle on commonly agreed-on approaches for fair multiclass classification and fair regression.
>
> - **Line 66, I may miss something, to compute the empirical risk minimization with dropping certain data points, the denominator in eq 2 should be the number of remaining data points?** – Observe that scaling the objective in Equation 2 by 1/ (N - #of dropped points), as you suggest, instead of 1 / N just changes the value of the objective but does not change the location of the optima. In either case, we recover the empirical risk minimizing solution.
>
> - **Line 152, the authors say that the weights for each data instance are restricted to be binary. Any motivations behind this restrictions?** – As mentioned in the paragraph preceding line 152, we have two primary motivations for this restriction --- computational efficiency and interpretability. In the binary case, $\mathbf{w}_{\mathrm{fair}}$ becomes available to us by just appropriately zeroing out coordinates of an N-dimensional vector of all 1s (Proposition 3.1). In contrast, without the binary constraint, we would need to run expensive iterative optimization procedures to recover w_fair. Moreover, binary w vectors are easy to interpret – we are either dropping or retaining training data instances. Without the binary constraint, such interpretations are harder to come by. For example, it is unclear what a negative weight ($w_n$) in Equation 2 implies.
>
> - **Line 2 in Algorithm 1: what does B stand for?** -- Thank you for catching this. B is the number of IHVP-WoodFisher iterations. We set this to 5000 in all our experiments. We will clarify this in the text.
>
>  - **Line 3 in Algorithm 1: the algorithm requires computing the fairness influence of each training data. But the definition of the fairness influence is from the true underlying data distribution and is computed in an expected manner. How do the authors compute these quantities given only access to the training data?** – These expectations are estimated from the training data via Monte Carlo approximations, under the standard assumption that the training data instances are sampled from the underlying data distribution.   See lines 73 and 97 of the file *influence_utils.py* in our submitted code.

---

### Meta-Review · Area_Chair_wY48 · 2022-08-26

**Recommendation:** Accept
**Confidence:** Less certain

**Metareview:**

This paper presents a tool for mitigating the influence of biased training data without retraining a model. The identified shortcomings of the paper include the restriction on binary classification setting. The meta reviewer recognizes this caveat but feels the focus aligns with the existing fairness definitions and the literature broadly. But the authors are encouraged to make it explicit about the paper’s limitations regarding the above restriction on binary classes and should clarify what and how the analysis and approaches can generalize to the multi-class case.

Prior to rebuttal, there was raised concern about the calculation of Hessian inverse and the generalizability to larger models. The authors addressed both during the rebuttal. Please add the new results to the final version.


**Award:**

No

---

### Decision · Program_Chairs · 2022-09-14

Accept